# Revealing the Corrosion Resistance of 316 L Stainless Steel by an In Situ Grown Nano Oxide Film

**DOI:** 10.3390/nano13030578

**Published:** 2023-01-31

**Authors:** Ying Ren, Yuchen Li, Jun Shen, Shaojun Wu, Liting Liu, Genshu Zhou

**Affiliations:** 1Center for Advancing Materials Performance from the Nanoscale (CAMP-Nano), State Key Laboratory for Mechanical Behavior of Materials, Xi’an Jiaotong University, Xi’an 710049, China; 2Analytical and Testing Center, Northwestern Polytechnical University, Xi’an 710060, China

**Keywords:** nano oxide film, repairability, long-term immersion, statistical method, 316 L stainless steel

## Abstract

It is widely accepted that the corrosion resistance of stainless steel originates from a compact Cr_2_O_3_ layer in the native passive film that serves as a barrier to aggressive ions. However, this suggestion has been questioned by some researchers. They believe that protectiveness might be related to the film recovery. Herein, the pitting development of bare 316 L stainless steel was compared with a corrosion-resistance enhanced steel obtained by tuning the native passive film of the alloy. Statistical software was employed for tracing the size and number of pits on the alloy surface. The statistical results for 12 weeks in 1 M sodium chloride solution (80 °C) revealed that there was a crossover in the growing rates of stable pits (diameter > 9 µm) between the bare alloy and the film-enhanced one. Stable pits on bare 316 L occurred early but showed a comparatively slow increase in the following weeks, demonstrating that self-repairability of metastable pits rather than impermeability of the native passive film plays the key role in the early stage of pitting corrosion.

## 1. Introduction

As normal steel tends to naturally return to its most stable form by way of corrosion, the emergence of stainless steel is bound to greatly delay the process [1]. Nowadays, the use of stainless steels, from daily necessities to chemical plants and sophisticated vehicles, is ubiquitous [2]. It is widely accepted that such extraordinary corrosion resistance comes from a native passive film on the alloy surface [3,4,5,6]. In particular, the formation of a continuous Cr_2_O_3_ layer in the 3-nm-thick native film after the addition of excessive Cr (>10.8 wt.%) was found to ensure the long-term protection of the alloy [3], enabling it to withstand harsh conditions. However, the fundamental mechanism of this protection is still unclear. In other words, the critical factor in localized corrosion (such as pitting, one of the most common and severe forms of corrosion), that is, film breakdown or pitting growth stability, has been debated for decades. Frankel et al. proposed that the protectiveness of the passive film plays an essential role under less aggressive conditions [7]. Most researchers tend to think that it should be related to the impermeability of the film [8,9,10,11]. This suggestion, on the one hand, is based on the efficient blocking of aggressive ions (e.g., chloride ions) by the compact Cr_2_O_3_ layer with fewer defects [8,12]. On the other hand, pitting only occurs after the protective film undergoes a breakdown event [9,11,13]. Some researchers, however, assume that the outstanding protectiveness might be due to the recovery (repassivation) of the oxide film after film breakdown [14,15,16]. Even if the stainless steel is scratched or otherwise damaged, the passive film instantaneously reforms whether in the air or in the water. Although both features are distinctive and essential for stainless steels, neither has direct experimental evidence for the superior protectiveness. For example, there has been a lack of a reliable method for quantitatively measuring the compactness of the native oxide layer. Meanwhile, the evaluation on the repairing ability of an oxide film on a metal surface, the repassivation potential, is closely related to the compactness of the native film (passive current density) [17].

The composition of the native film is primarily determined by the alloy and the recovery process is also affected by the electrolyte associated with the dissolution of a locally exposed alloy. This tangled relationship together with a small dimension of film breakdown is bewildering, limiting our understanding of the specific roles of impermeability and repairability in the initial stage of pitting corrosion [14,18,19,20]. Considering the complex dynamic interplay of multiple factors, including alloy, film, and electrolyte [21,22], we propose to introduce a modified native passive film on the alloy surface for comparison with bare alloy in pitting corrosion. We have reported a superior anticorrosive coating (~8 µm in thickness) grown in situ on a Mg–Li alloy using a mild low-temperature plasma [23]. The same method could also be applied to inert metals and alloys where a several-nanometer-thick metal oxide film was found to form on the metal surface. The thickness of the oxide film on 316 L SS varies from 5 to 15 nm depending on treatment time [24]. The film was similar to the native film in chemical composition, crystalline structure (amorphous), and kinetic growth, but exhibited a tunable enhanced corrosion resistance due to chemical ordering (well-defined chemical layered structure) and improved film quality. The thickened continuous Cr_x_O_y_ layer (2–3 nm) in the film plays the key role in corrosion resistance, which is stable as an ultrathin barrier for the underlying 316 L alloy. It could decrease the anodic current density of bare alloy by two orders of magnitude with a remarkable increase in pitting potential in 3.5 wt.% NaCl solution. The enhanced protectiveness for the alloy was also observed under salt spray for six months [24]. These features make it possible to conduct long-term environmental experiments. Moreover, this film enhanced (FE) alloy experienced the same stages of pitting development as the bare alloy, that is, film breakdown, metastable pit growth (small pits that cease to grow at this stage), and stable pit growth (pits that can grow quite large) [25]. As the modified oxide film is relatively compatible with the native one in thickness and chemical composition, its breakdown could be repaired by the native one. This, however, could not be observed on dissimilar-material or traditional-coating enhanced alloy surfaces because of accelerated stable pitting corrosion due to surface electrochemical heterogeneity [26]. Thus, the development of pitting corrosion of the FE alloy can be compared with that of the bare one with a native film. This FE alloy, firstly, enables one to exclude the influence of alloy composition due to the same substrate. Second, unlike micrometer-thick coatings in which the chemical composition of electrolyte drastically changes in deep pits [25], such a metal oxide film with a thickness (5 to 15 nm) close to the native one (2 to 3 nm) offers an open environment for the stabilization of electrolyte when film breakdown occurs.

As the film breakdown and recovery are stochastic in time and space [27], it is difficult to draw a line of demarcation between the two stages during the initiation of pitting corrosion. The problem, thus, can possibly be solved via statistical tools. It is noteworthy that traditional statistical results are often provided by electrochemical methods and fail to describe long-term process of pitting [17,18,28,29]. In this respect, we use statistical software to obtain statistics on pit size distribution. The next step consists in analyzing the variation in pit number and size resulting from a long-term immersion time to establish the dominant factor of the bare alloy during the protection. More importantly, this factor can be distinguished by comparing the bare alloy with the FE one. The development of stable pits was considered as the evaluation criterion because it was the principal cause of corrosion failure (without considering other corrosion forms) [25]. As illustrated in Figure 1a, ideally, the initial stage (stage one) for impermeability-dominated protection from surface oxide film is pit free. If the number of both metastable and stable pits on the bare alloy is greater than that on the FE alloy (I) in the following stages (stage two and stage three), it suggests that the corrosion resistance of 316 L SS is impermeability dominated. This is because the recovered native film very weakly suppresses the stabilization of metastable pits (I and II). Otherwise, if the number of metastable pits on the bare alloy at the initial stages (stage one and stage two) exceeds that of the FE alloy and the quantity of stable pits in the following stage (stage three) is not higher than that of the FE one, the corrosion resistance is assumed to be self-repairability-dominated (I and III). This means that the growth rate of stable pits on the bare alloy is much lower than that of the FE alloy due to a self-repairing property, and consequently there must be a crossover in the development of stable pits between the two (Figure 1b). For self-repairability-dominated protection (III), the film breakdown and metastable pitting occur early (see stage one in Figure 1a) because of the weak impermeability of the native passive film. However, stable pits develop slowly for this protection because few small pits can grow into stable large pits due to excellent self-repairability of the alloy surface [20].

## 2. Materials and Methods

### 2.1. Materials

316 L stainless steel (69.27 wt.% Fe, 16.38 wt.% Cr, 10.69 wt.% Ni, 2 wt.% Mo) was purchased from China Baosteel Co., Ltd. (Shanghai, China). Samples were cut from an industrial annealed 3 mm-thick plate. The specimens were ground and polished using 600 grit polishing paper followed by a 0.5 µm diamond paste polish to avoid the effects from surface toughness or a deformation layer [30]. They were then cleaned with acetone in an ultrasonic bath and subsequently cleaned with pure alcohol and distilled water. After cleaning, the samples were dried under compressed air. The FE alloy was fabricated by a dielectric barrier discharge (DBD) system. The DBD system is considered as a post-discharge bipolar ions source in industry. Oxygen-bearing ions were obtained by applying a sinusoidal voltage between two parallel electrodes covered by silica glass. The input voltage was 30 to 45 V, and the input current was 1.3 to 2.0 A. In air at atmospheric pressure, discharge filaments are homogeneously distributed on the dielectric surface. The ambient atmosphere with humidity ranging from 20% to 45% was thereby introduced into the system at room temperature. Since this low-temperature surface treatment was mild and the corrosion resistance of an alloy increases with treatment time, 10 h was applied to grow the oxide film on alloy surface in situ. The temperatures of the metal surfaces during processing were 80–110 °C. More technical details can be found in Ref. [24].

### 2.2. Characterization

For the electrochemical measurements, samples with areas of 1 cm^2^ were used as the working electrodes, a Pt mesh was served as the counter electrode and saturated calomel was taken as the reference electrode. The electrochemical measurements were carried out using a potentiostat (VersaSTAT3F, Princeton, Oak Ridge, TN, USA) with a voltage sweep rate of 2 mV per second without deaerating. Polarization was applied after ~15 min at open circuit potential (OCP). The reverse potential was determined by the anodic current density that was three orders of magnitude higher than that at the pitting potential for bare 316 L SS. The measurements were performed independently in triplicate. For the immersion experiments, the sample was immerged at 80 °C in 1 M NaCl solution made with reagent grade chemicals in high pure (18 MΩ resistance) water. The sample was placed on a homemade wooden support to avoid crevice corrosion. The optical microscopy images of the corrosion morphology of the 316 L samples were acquired by a laser scanning confocal microscope (OLS4000, Olympus, Tokyo, Japan) in rapid imaging mode. Nano Measurer software was used to count pits and measure their size in the high-resolution optical images. Prior to the auto identification, all pits had to be marked in the images.

The high-resolution transmission electron microscopy (HRTEM) was recorded using a TEM (JEM-F200, JEOL, Peabody, MA, USA) operating at 200 kV. Samples for cross-sectional imaging were prepared via a focused-ion-beam (FIB) lift-out technique using a Helios 600 FIB-SEM setup (FEI, Hillsboro, OR, USA), and a carbon over-layer was deposited on the FE alloy surface via an electron beam to protect the ultra-thin oxide film. After thinning the total thickness of the sample down to 100 nm, the specimen was Ar ion milled to an electron transparent specimen starting from 5 keV down to 2 keV as the final cleaning energy.

## 3. Results

### Structure and Chemical Composition of Films

Figure 2a displays the cross-sectional high-resolution TEM image of the FE alloy. The increased thickness (~7.9 nm, and the native oxide film on 316 L SS is ~3 nm) and amorphous structure on the (111) plane revealed by fast Fourier transform spectrum could be observed. The detailed characterization of the composition and structure of the thickened oxide film can be found in Ref [24]. The dynamic polarization curves revealed that the reverse scan for the FE alloy had returned exactly to the repassive potential of the native film (~−1.4 V) on the bare alloy, indicating that the repairability of the film was very weak and that the film can only be repaired by the native film. The enhanced protectiveness and long-term stability of the FE alloy were shown to be owing to a thickened continuous Cr_2_O_3_ layer (two to three nanometers) as well as the improved film quality [24]. Therefore, the corrosion resistance of the FE alloy was a typical impermeability-dominated protection. Such protection from robust ultrathin barriers or two-dimensional materials, such as graphene and borate nitride, has been reported recently [31,32,33]. However, these dissimilar materials would skip the metastable pitting and cause severe accelerated stable pitting once film breakdown occurs [26], which makes them unsuitable for studies of pitting comparison.

To observe the development of pitting, the samples were immersed in 1 M sodium chloride solution at 80 °C. An area of 1.2 × 1.2 mm^2^ was selected to obtain accurate statistics, as shown in Figure 3. After 3 weeks, several stable pits (>9 µm in diameter) appeared on the bare alloy, and plenty of small metastable pits could also be observed. In contrast, stable pits on the surface of the FE alloy were hardly observed, and small metastable pits were also very scarce. The statistical results reveal that most of metastable pits were less than 3 µm in size, and the number of 1 to 3 µm pits on the bare alloy was about 8 times that on the FE alloy. After 6 weeks, several stable pits appeared on the FE alloy, and the number of metastable pits increased rapidly in this stage. While the quantity difference between the metastable pits in the two samples decreased, the number of stable pits on the bare alloy still exceeded that on the FE alloy. After 12 weeks, unexpectedly, the two samples were corroded to nearly the same degree. The statistical results revealed that the quantity of both metastable and stable pits on the FE alloy had increased more than ten times over the previous six weeks, which had far exceeded the increase on the bare alloy. Moreover, the newly formed stable pits on the bare alloy were few. These results clearly demonstrate that the anticorrosive oxide film could protect the alloy at the beginning stage of film breakdown (the first two stages). However, the protection was not so promising when film breakdown occurred frequently and it could not be well repaired. By comparison, the native film on 316 L SS could slow down the development of both metastable and stable pits in the following weeks, revealing the key role of self-repairability in long-term corrosion resistance of 316 L SS. It should also be noted that the development trends of pitting corrosion summarized in present study might not be applicable to other metals and alloys. As the chemical and structural stability of native passive film is dependent on the electrolyte, the repairing ability related to alloy composition is also influenced by the electrolyte, and it may degrade and lead to rapid growth of stable pits in hash conditions. Thus, pitting development is dependent on both alloy composition and electrolyte.

Some other important conclusions can also be drawn from the statistical results. First, to be precise, the critical size of stable pits on 316 L SS was ~9 µm rather than 10 µm [18,34]. Moreover, the 7 to 9 µm pits were the fewest and their growth was also the slowest. This was especially noticeable for the FE alloy where no such pits were observed during the long-term immersion. Note that the pits smaller than 1 µm were not counted in the total number of metastable pits due to the decrease in the statistical accuracy resulting from the limited image resolution. The growing rates of both metastable and stable pits are shown in Figure 3d. The metastable pits in both samples rapidly grew with immersion time, but their growth rate in the FE alloy at the initial stage was obviously slower than that in the bare alloy due to the excellent impermeability of the thickened oxide film (Figure 2). As for stable pits, the growing rate was slow on the bare alloy, but it was extremely low at the initial stage and then accelerated on the FE alloy, and a crossover could thus be observed in the later period between the two samples. These results conformed exactly to our models of self-repairability-dominated protection III and impermeability-controlled protection I in Figure 1b. For the former model, the metastable-to-stable transformation rate should decrease with time due to weak impermeability but strong self-repairability, as in case of bare 316 L SS in Figure 3e. Considering the latter model, no such trend was observed on the FE alloy. In addition, these two protection mechanisms also differed in the growth rate of pit size. As shown in Figure 3e, the rate of transformation of 1 to 3 µm pits to 3 to 7 µm pits for both samples increased with time, but the growth rate on the bare alloy was much faster than that on the FE alloy. This suggested another feature of self-repairability, that is, sacrificing growth to avoid abrupt stabilization.

Earlier studies found that the sudden drop in the anodic current density of a metastable pit on stainless steels occurred over timescales of 10 milliseconds to several seconds, which was attributed to the oxidation of Cr on the alloy surface [35,36,37]. Hence, it was difficult to directly observe the formation of this repassivation. Recently, Xie et al. proposed a percolation process to account for the initial stage of the repassivation based on density functional theory. Isolated -Cr-O-Cr- mer units were suggested to result from selective dissolution of active metals [14]. The nucleation and reproduction of metastable pits have been investigated by many researchers via potentiostatic polarization [17,18], potentiostatic polarization [20] and electrochemical noise [38,39]. It was shown that the increase in corrosion current density during this stage represented an explosive growth in the number of metastable pits rather than the stabilization of individual stable pits. However, the size change of metastable pits is often ignored because of lacking long-term studies on pitting corrosion of stainless steels. Obviously, the increase in the size with immersion time for the bare 316 L SS (Figure 3e), considering their comparatively high density and large diameters (1 to 7 µm, close to stable pits), should also contribute to a significant portion of the corrosion current density. The growth of metastable pits is preceded by frequent activation and repassivation events [20]. Thus, the relatively rapid increase in size in the following weeks but an extremely slow rate of transformation to stable pits (Figure 3e), compared with that of FE alloy, probably indicates a high-frequency occurrence of such events as a result of excellent self-repairability of 316 L SS during this stage. Moreover, this high frequency together with the reproduction of pits helps disperse the anodic current density of the alloy surface at the open-circuit potential, and greatly reduces the probability of the stabilization of metastable pits.

It should also be noted that enhanced corrosion resistance at the initial stage was not observed in the 6% FeCl_3_ solution, as the oxide film could not survive under such a harsh condition. It was, therefore, assumed that the recovery of the native film would also be impossible, thus the pitting might have rapidly initiated from defects or inclusion sites on the alloy surface in a stabilized way [40]. Meanwhile, the main corrosive behavior changed from the film-related 2-D surface to the bulk-related 1-D hole (see Figure 4), showing a shift to the aggravation of electrochemical heterogeneity, which is irreversible. Therefore, for a superior anti-corrosive alloy, if its native film enables stabilization in a less aggressive electrolyte, the film would be able to resist the ingress of aggressive ions. If the film undergoes a breakdown or a scratch, the rapid repair achieved by forming a dense and inert oxide layer on the alloy surface that is similar to the native film in composition and structure to avoid surface electrochemical heterogeneity, is a more reliable strategy in the long run.

## 4. Conclusions

By avoiding influences from alloy and local chemical variations, an in situ grown anticorrosive ultrathin oxide film enabled clarification of the role of native passive film on 316 L SS during the initiation of pitting corrosion in 1 M NaCl solution (80 °C). A comparative analysis of the statistical results on pits in terms of number and size with immersion time revealed that there was a crossover in the growing rates of stable pits between the bare alloy and the film-enhanced alloy. For the bare alloy, the stabilization of some metastable pits appeared early, but their development rate increased very slowly. In contrast, stable pits could not be observed for a long period of time due to the anti-corrosive oxide film, but were manifested in the following stage. Because of its intrinsic self-repairability, it is demonstrated that the recovery of native passive film plays the key role in suppressing the stabilization of metastable pits and thereby ensures the extraordinary long-term protection for the alloy.

## Figures and Tables

**Figure 1 nanomaterials-13-00578-f001:**
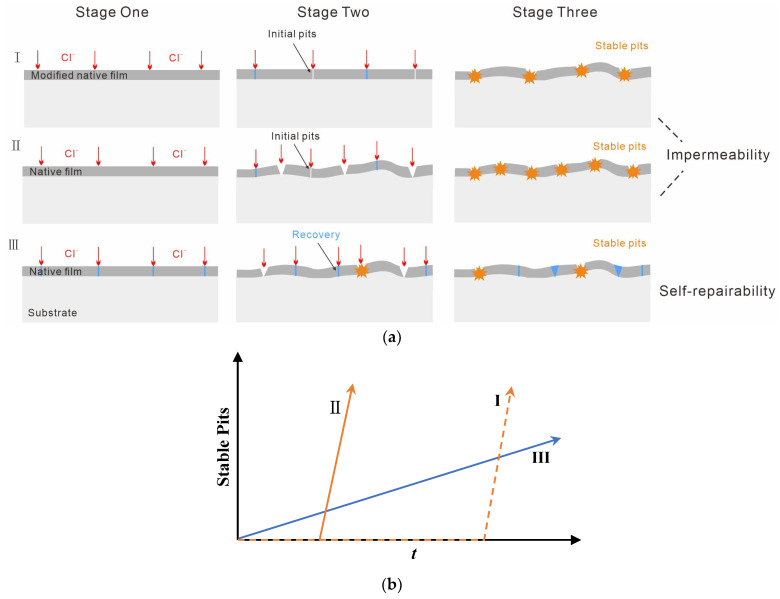
(**a**) Illustration of the determining dominant factors comparing the development of pitting: I, FE alloy with a typical impermeability-dominated corrosion resistance exhibits the least number of metastable pits at the initial stage due to the modified oxide film; II, Bare alloy with impermeability-dominated corrosion resistance, for which the native film can offer protection for a period; III, Bare alloy with a self-repairability-dominated corrosion resistance, in which metastable pitting occurs early but the stabilization of pits needs a long period of time. (**b**) The development of stable pits for each protection mechanism.

**Figure 2 nanomaterials-13-00578-f002:**
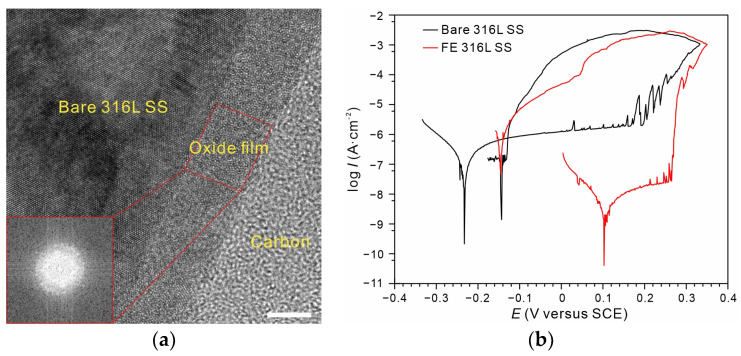
(**a**) Cross-sectional HR-TEM image of the modified oxide film on 316 L SS (Scale bar, 5 nm). (**b**) Polarization curves of bare 316 L SS (black) and FE 316 L SS (red) with a reverse scan in 0.6 M NaCl solution (25 °C).

**Figure 3 nanomaterials-13-00578-f003:**
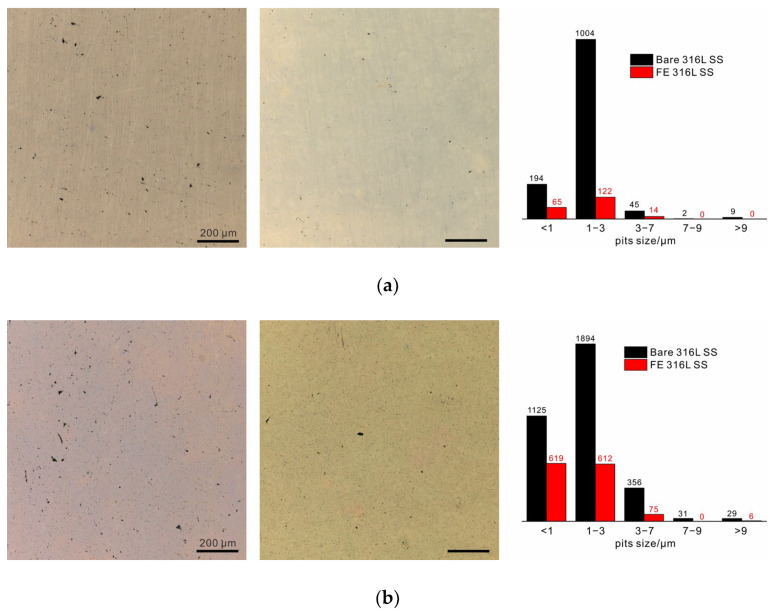
Optical microscopy photographs of 316 L SS (left) and FE 316 L SS (right) after immersion for (**a**) 3 weeks, (**b**) 6 weeks, and (**c**) 12 weeks, and the corresponding statistical results. (**d**) The development of metastable pits and stable pits, respectively. (**e**) The metastable-to-stable pit transformation ratio and the transformation ratio of metastable pits with small sizes (1–3 µm) to metastable pits with larger sizes (3–7 µm).

**Figure 4 nanomaterials-13-00578-f004:**
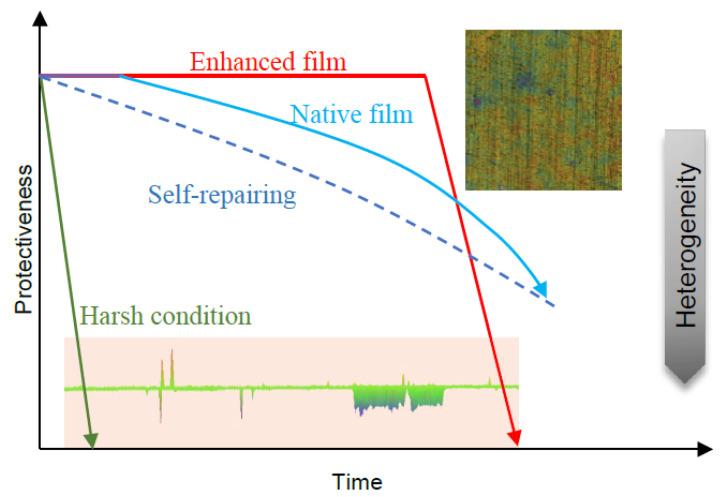
Schematic of critical step for three conditions of pitting: native film without protection under harsh condition (green), native film with excellent self-repairing properties (blue), and the corrosion-resistance enhanced ultrathin oxide film with weak self-repairing properties (red).

## Data Availability

The datasets used and analyzed in the current study are available from the corresponding author on reasonable request.

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
