# Peer review of "Revealing the Corrosion Resistance of 316 L Stainless Steel by an In Situ Grown Nano Oxide Film"

_nanomaterials, 2023, doi:10.3390/nano13030578_

Round 1

Reviewer 1 Report

The authors demonstrated a competition between growing of stable pits and in situ grown ultrathin oxide film (which is corrosion protective) during long-term immersion of 316L stainless steel in 1 M NaCl solution (80℃). The enhanced film may lead to reparability in initial stage of pitting corrosion. However, some correction should be made:

-Abstract needs to be completed by emphasizing the applied techniques and statistic method, and experimental results

-The phrase ‘The FE alloy was fabricated as described in Ref [19]’ (lines 111,112) is insufficient. More details about metal surface processing are needed.

-Line 137: write ‘The potentiodynamic polarization curves revealed…’

-Check the electrode potential of -1.5 V (which is not appeared in Figure 2b)

-Line 211: write ‘Recently, Xie et al. proposed…’

-Line 215: write separately the references using potentiodynamic (not potentiostatic) polarization) [ ] and electrochemical noise [ ]

- Authors should correct the references according to journal requirements. Please add also DOIs of the references.

Reviewer 2 Report

Comments to the Authors:
The authors of this paper present a statistical analysis on pit size distribution and then an analysis of the variation of pit number and size with the long-term immersion time to establish the dominant factor of the bare alloy during the protection. However, some details should be considered by the authors:

GENERAL COMMENT: Please check the punctuation throughout the text

COMMENT: Page 1, line 26: More recent references could be added.

COMMENT: Page 2, lines 90-94: “It means … protection (III).” More comments (and references) could be added by the authors.

COMMENT: Page 4, line 147: “might not be applicable to other metals and alloys.”. Why? More comments/discussion could be added.

The results support the conclusions of the authors and I think that this paper may be published.

Reviewer 3 Report

My remarks are included in a seprated file.

Author Response

Many thanks for the reviewer's suggestion.